# Changes in Nutrient Declaration after the Food Labeling and Advertising Law in Chile: A Longitudinal Approach

**DOI:** 10.3390/nu12082371

**Published:** 2020-08-08

**Authors:** Daiana Quintiliano Scarpelli, Anna Christina Pinheiro Fernandes, Lorena Rodriguez Osiac, Tito Pizarro Quevedo

**Affiliations:** 1Carrera de Nutrición y Dietética, Facultad de Medicina-Clínica Alemana, Universidad del Desarrollo, Santiago 7610658, Chile; d.quintiliano@udd.cl; 2Escuela de Salud Pública, Universidad de Chile, Santiago 7910000, Chile; lorenarodriguez@med.uchile.cl; 3Facultad de Ciencias Médicas, Universidad de Santiago, Santiago 7940702, Chile; tito.pizarro@usach.cl

**Keywords:** food policy, obesity prevention, front-of-package labels, warning label, energy and nutrients of concern declaration, Chile

## Abstract

Chile has implemented several strategies to decrease the burden of obesity and chronic diseases. The Food Labeling and Advertising Law (Law 20.606) requires a front-of-package “high in” warning label when energy and nutrients of concern (ENC) (total sugar, saturated fats, sodium) exceed established limits. This study aims to evaluate the impact of Law 20.606 on the ENC declaration of packaged foods in Chile, before and after the law implementation. We analyzed food nutritional labeling declarations from 70% of the most consumed packaged foods in Chile. Data collection was conducted in 2013 and 2019 in Santiago. Pictures from all sides of the package were taken from 476 products, classified into 16 food groups. All food groups had changes in the ENC declaration during the study period. Total sugar content showed the highest reduction (−15.0%; *p* = 0.001). Dairy, confitures and similar and sugary beverages had the greatest reduction in energy and total sugar content (*p* < 0.01). Energy, total sugar and sodium front of package “high in” simulation was significantly reduced in dairy, sugary beverages, flour-based foods, confitures and similar, fish and seafoods, fats and oils, spices, condiments and sauces and sugars (*p* < 0.05). We observed that companies reformulated products to adapt to the new regulation.

## 1. Introduction

The most prevalent causes of mortality and morbidity in Chile relate to non-communicable diseases (NCD), with one third of the population classified as obese [1]. The high consumption of energy and nutrients of concern based on foods high in sugars, sodium and saturated fats is the most important risk factor associated with diet-related NCD [2]. Furthermore, 28.6% of total energy intake of the Chilean diet is derived from ultra-processed foods [3].

International organizations, such as the World Health Organization (WHO), the Pan American Health Organization (PAHO) and the Organization for Economic Co-operation and Development (OECD) have encouraged countries to prioritize prevention of overweight and obesity in public health polices, calling for policies focused on market regulation and policies to revert tendencies of increased consumption of unhealthy foods. In addition, policies on the restriction of sales of processed foods, sugary drinks and the poor nutritional content of fast food and its relation to NCD are encouraged [4,5,6].

In this context, Chile has implemented several strategies to decrease the burden of obesity and chronic diseases, including specific laws, plans, and programs [7]. A pioneer in the region, one of the most important was the Food Labeling and Advertising Law (Law 20.606); An innovative policy that implemented a nutritional profile limit for energy and nutrients of concern (total sugars, saturated fats, sodium) for incorporating a front-of-package (FoP) “high in” warning label. Moreover, this same regulation restricts the marketing of “high in” foods to children under 14y and prohibited the sale of these products at schools. The implementation of the Law 20.606 occurred in three periods: 2016 (less restrictive limits), 2018 (more restrictive limits) and 2019 (final limits for energy and nutrients of concern). The primary focus of this regulation was to inform consumers about the high content of energy, total sugar, saturated fat and sodium in processed foods and to improve the food environment for children, restricting their exposure to excessive advertising and sales of unhealthy foods [8,9]. Another objective of this regulation was to support the reformulation of processed foods, decreasing total sugars, saturated fats and sodium content or replacing ingredients that improve the nutrient profile.

The presence of FoP warning labels on products improve consumer comprehension of nutritional facts, help consumers identify products high in nutrients of concern, and discourage consumers from purchasing these products [10,11]. An Australian study showed that FoP labels could reduce intended sugar-sweetened beverages purchases independent of type, with the magnitude of effect greatest for graphic warning labels (OR: 0.22; 95% CI: 0.14–0.35) [12].

Researchers compared the Chilean FoP warning label with the traffic light system among children in grades 4 to 6, and showed a positive effect of the Chilean model in the choices of processed foods (wafer cookies and orange juice), demonstrating a possible impact on the consumption of nutrients of concern [13]. The availability of unhealthy foods in school kiosks in the capital, Santiago, was analyzed in 2014 and six months before the law came into force in 2016. The authors found a reduction in availability of 75.4% of these products and a reduction in energy, total sugar, saturated fat and sodium in the nutrient content of solid products [14], demonstrating an overall positive outcome of the Law, related to an improvement of the school food environment.

Few studies have established the relationship between FoP warning label implementation and product reformulation. In a simulation study, Ares and cols (2018) found that FoP warning labels could improve consumer intention to select more healthful foods, decreasing intake of nutrients of concern [15]. Another simulation study investigated consumer willingness to pay for a reformulated product and found that this condition was affected by the brand type and sodium and fat reductions in sausages samples [16]. These results demonstrate that implementation of a FoP warning label system could be followed by a reduction in consumption of unhealthy foods after reformulation, which, in turn, could reduce the exposure to this diet-related risk factor for NCD.

Kanter et al. (2016) demonstrated a minimal impact of product reformulation before Law 20.606 implementation in Chile, with few reductions in energy content and increases in some nutrients of concern by food groups [17]. There have been no studies considering the latest implementation period (June 2019) of the law. Thus, the purpose of this study was to evaluate the impact of Law 20.606 in energy, total sugar, saturated fat and sodium declaration of packaged foods commercialized in Chile, comparing nutrient content information prior to the law implementation (2013) and after the last implementation stage (2019). We hypothesized that companies reformulated products to adapt to the new regulation, reducing energy and the content of nutrients of concern.

## 2. Materials and Methods

### 2.1. Study Design, Sampling and Data Collection

We conducted a descriptive, observational study that analyzed food nutritional labeling information from the most consumed packaged foods in Chile. Data collection was performed in two periods, in October–November 2013, before Law 20.606 was implemented and in July–September 2019, after the last stage. We selected 70% of the most consumed packaged foods in Chile, identified in the 24 h recall database (total sample of 4920 interviews and 844 foods) of the National Food Consumption Survey (2009–2010) (ENCA) [18].

Data collection was carried out in the biggest supermarket chain in Santiago, Chile, after a formal agreement was reached between the Chilean National Association of Supermarkets (ASACH) and the Universidad del Desarrollo (UDD). A set of pictures of all sides of the selected foods packages was taken by trained fieldworkers with a high-resolution camera (EOS Rebel T100, Canon^®^, Santiago, Chile). In 2013, pictures of ~650 products (~3000 pictures) were taken. After picture quality was evaluated (sharpness of pictures), ~500 pictures from ~100 products were excluded from this database.

In 2019, the food list database and pictures from 2013 were recovered (551 products, ~2500 pictures) and used as a baseline database for data collection. First, all products were searched in the same supermarket chain; the ones not found were searched in other supermarkets chains or local markets. In sum, 13.6% of the products were not founded in Santiago in 2019. The total sample and product losses in the period are described in Table 1.

The pictures database (2013–2019) was built and managed using REDCap (Research Electronic Data Capture), hosted by UDD. All nutritional labeling information, product name, brand, manufacturer, and ingredients were registered by trained personnel.

### 2.2. Food Groups Classification

Products were classified in food groups according to the Chilean Sanitary Regulation Food Code [8]. The total sample was classified into 16 food groups: dairy (e.g., milk, cheeses, butter, yogurts, milk pudding, and desserts); flour-based foods (e.g., cookies, crackers, cakes, bread, semolina, noodles, and breakfast cereals); confectionery and similar (e.g., chocolate bars, candies, and jellies); confitures and similar (e.g., compotes and jams); fish and seafood (e.g., tuna, clams, mussels, oyster); sauces and dehydrated soups (e.g., seasoning sauces, bases, powder soups, and creams); meat and derivatives (e.g., ham, sausages, chicken and turkey marinade meat, pates); sugary beverages (e.g., sugary juices, drinks, and sodas); cereals (e.g., cereal bars, flavored oats, rice); foods for special diets (e.g., babies formula and baby foods); fats and oils (margarine, vegetable oils, whip cream); desserts and ice creams; spices, condiments, and sauces (e.g., tomato sauces, ketchup, other salty sauces); sugar (e.g., refined and brown sugars, flavor powders, sugar sauces); canned foods (e.g., fruits and vegetables); prepared meals (pasta and ready to eat noodles) (Table 1).

### 2.3. Simulation of “High in” FoP Warning Label

To analyze the energy and nutrients of concern (total sugars, saturated fats, sodium) declaration, the limits for nutrient profiles from the final stage of the Chilean Food Law were considered [8].

When nutrient content per 100 g is greater than or equal to the established limit, the FoP “high in” warning label is applied to packaged foods. The limits established for solid foods per 100 g are: 275 calories; 10 g total sugar; 4 g saturated fat; and 400 mg sodium. For liquid foods, the limits established per 100 mL are: 70 calories; 5 g total sugar; 3 g saturated fat; and 100 mg sodium. To simulate the presence/absence of FoP (2013 and 2019), following the law guidelines [8], only products with nutrients of concern added were included in the analysis. This process was conducted by a trained professional that identified the possibility of containing added sugar, sodium or saturated fats or other caloric nutrient based on the type of product and the ingredients listed on each food package.

### 2.4. Statistical Analysis

Numeric variables (energy, sugar, saturated fat, and sodium content) were presented as median and interquartile rank, considering that they had a non-parametric distribution according to the Shapiro-Wilk test. The comparison of the magnitude of the differences in nutritional labeling information (2013–2019) for these nutrients was evaluated using the non-parametric paired Wilcoxon signed-rank test.

The simulated FoP “high in” warning labels were presented in absolute numbers, proportions, and 95% confidence interval according to food groups, type of product (solid or liquid) and year. The difference in warning label proportion for each food group during the studied period was analyzed by Chi-Square or Fisher’s Exact Test (when more than 20% of the observations had *n* < 5).

The level of statistical significance considered for all analyses was *p* < 0.05. Stata 16.1 software (Texas, USA) was used.

## 3. Results

In 2013, a total of 551 products were included in the pictures database. In the follow-up period, 75 products were not found (13.6%) despite the exhaustive search in different supermarket chains and local stores. This means that these products were not present in the market for different reasons like seasonality or discontinuation (Table 1). The nutritional content (energy, *p* = 0.558; total sugar, *p* = 0.471; saturated fat, *p* = 0.507; sodium, *p* = 0.662, data not shown) and FoP simulation (energy, *p* = 0.069; total sugar, *p* = 0.660; saturated fat, *p* = 0.324; sodium, *p* = 0.067, data not shown) of these products did not differ significantly from the found ones, but they were excluded from these analyses. In 2019, 450 products of the baseline database were located, and 16 products were replaced for another product with similar characteristics, such as type, brand, manufacturer (e.g., berry-flavored yogurt was changed for strawberry-flavored yogurt).

In the general simulation of the FoP warning label, we observed that total sugar was the nutrient of concern that showed the highest reduction during the period (−15.0%; *p* = 0.001), followed by sodium (−9.2%; *p* = 0.118), energy (−3.9%, *p* = 0.388), and saturated fat (−1.5%; *p* = 0.861) (Figure 1).

### 3.1. Changes in Nutrients Declaration for Solid and Liquid Products (2013–2019)

All food groups had some changes in energy and nutrients of concern declarations during the study period. Table 2 describes the nutritional labeling of solid foods. All food groups had a reduction in the energy level and significant changes were observed in three groups (flour-based foods, confitures and similar, fats and oils). The confitures and similar group had the highest median reduction, with almost 50.0%. For total sugar, significant reductions were observed in dairy, confitures and similar and the cereals group, with −62.5%, −60.0% and −21.7%, respectively (*p* < 0.05). Saturated fat did not show significant changes in the period. Sodium content was significantly reduced in three groups, −40.8% and −37.5% for fats and oils and spices, condiments, and sauces groups, respectively. Dairy and meat and derivatives groups had an increase of 4.2% and 13.0% (*p* < 0.05).

Table 3 presents data for liquid products. A reduction in energy was observed for the dairy and sugary beverage groups, −17.2% and −52.2%, respectively (*p* < 0.01). Moreover, in dairy and sugary beverages a considerable reduction in total sugar was also observed (−50.0%; −63.6%, respectively, *p* < 0.01). Saturated fat content did not have any significant changes in the food groups. Finally, sodium content presented a minor reduction in the dairy group and a 40.0% increase in the desserts and ice creams group, but these changes were below the limits established in the food law (100 mg/100 mL) (*p* < 0.05).

### 3.2. Simulation of FoP Warning Label Presence on Solid and Liquid Products (2013–2019)

Table 4 presents the simulation of “high in” FoP warning labels for solid products according to year. It was observed that just two groups had a significant reduction in the proportion of FoP “high in” for energy (flour-based foods and fats and oils). The other groups did not experience significant changes in the proportion of possible “high in” energy.

For the “high in” total sugar simulation, three food groups presented a significant reduction in the proportion of labels group. A significant increase in the proportion of FoP labels for this nutrient of concern was observed in spices, condiments and sauces and food for special diet group. Despite not observing a statistically significant difference, the dairy, cereals and canned foods groups showed an important reduction in the proportion of FoP for this nutrient. For saturated fat, two food groups had a significant reduction on the proportion of the FoP “high in” (fats and oils and meat and derivatives). The sauces and dehydrated soups group decreased by almost −12% the FoP in 2019, but this change was not statistically significant.

“High in” sodium had a strong reduction during the study period for the following three food groups: fish and seafood, fats and oils, and spices, condiments and sauces, emphasizing that 100% of flour-based foods absented from this label in 2019 (all *p* < 0.01). For the prepared meals group, an increase in the proportion of this “high in” label was observed. The sauces and dehydrated soups group did not experience any changes in the period (Table 4).

For liquids (Table 5), the fats and oils and desserts and ice creams groups did not experience significant changes in the proportion of FoP labels for any nutrient of concern. On the other hand, the presence of energy, total sugar, and sodium FoP “high in” was significantly reduced in the dairy group, while saturated fat significantly increased from 16.3% to 22.5% (*p* < 0.05). For the sugary beverages group, an important reduction in the presence of FoP “high in” label for energy (8.7% to 4.3%) and total sugars (66.7% to 17.4%) was observed.

## 4. Discussion

This impact evaluation of the Chilean Food Labeling and Advertising Law on nutrient declaration showed an extensive decrease in energy, total sugar and sodium content for the most consumed packaged food products (period 2013–2019). The highest reduction was observed in total sugar content, while few changes were observed in saturated fat content. This could indicate an industry reformulation on some products groups during these period. Our analysis was based on food labels declaration and in Chile, compliance with food label regulation is around 70% to 80% [19].

An important effect found in our study was that the FoP “high in” labeling implemented in Chile has induced a change in nutrient declarations of products, in food groups highly consumed by the population and especially among children, such as cereals, dairy and sugary beverages. On the other hand, very little changes were observed in pastry and desserts and ice creams products, in which reformulation is more complex. These changes represent positive advances in creating a healthier food environment. In addition, it is important to remember that the most relevant objective of FoP “high in” labels is to provide clear, simple and truthful information regarding the nutritional quality of food, so that consumers can make informed purchases [20].

The first evaluations of the impact of the Law 20.606 (2018) showed that the population recognizes, supports, and understands the regulation and around 50% are using FoP “high in” labels when purchasing foods. Moreover, according to food-producing companies, food reformulation has occurred in about 18% of package foods [21]. Kanter et al. (2019) analyzed changes in ENC declaration of packaged food in Chile in 2015 and 2016 (*n* = 5241 and *n* = 5479) and found few changes (less than 5%), reinforcing that there was minimal reformulation prior to the implementation of the Law [17]. Nonetheless and considering this minimal reformulation, these changes in nutrient content are reflected in the fact that many products that would have had a FoP “high in” label for energy and nutrients of concern, when food law came into effect (June 2016) did not have one.

With the implementation of the law occurring in stages and the final change happening in 2019, the results of our study imply that consumers can now find products with fewer FoP “high in” warning labels and, as a result, be exposed to lower levels of energy, total sugar, saturated fat and sodium in some food groups such as, dairy, sugary beverages, and flour-based foods. It is important to mention that food producers that decrease the content of nutrients of concern can avoid marketing restrictions and continue with sales in school environments, making reformulation very attractive for food industry.

A review of experimental studies of FoP warning labels on sugary beverages and processed food conclude that the strategy was easy to understand and allowed consumers to identify and purchase foods with a lower content of energy and nutrients of concern, which, in turn, helps consumers to rank healthier products [11]. Other models of FoP warning labels implemented in Holland (“Choices”) and in New Zealand (“Health Star Rating”) have been associated with a decrease in sodium content, saturated fat, added sugar, energy and an increase in fiber content on reformulated and in new products [22,23].

According to our data, dairy presented a high decrease in total sugars (liquid and solid) and energy (liquid). These results could impact the intake of energy and nutrients of concern among children, considering that these products are highly consumed among this age group [18]. One limitation in the dairy group is that each food producer is allowed to define the consistency of the product as either a solid or liquid [8] and indeed this could mask compliance with the regulation. Massri et al. (2018), observed that in the period after implementation of Law 20.606 this food group increased in availability, with respect to total products, in school kiosks of Santiago (1.7% in 2014 to 5.4% in 2016) [14]. This could be mainly due to a reformulation process that was going on in this period and changes in its food matrix, principally focused on decreasing total sugar content and replacing sugars to non-caloric sweeteners. With this strategy, products could be sold at school kiosks since they did not have any FoP “high in” labels. Dairy are widely consumed by preschool and schoolchildren in Chile (e.g., milk, cheese, yogurt, and milk-based desserts) complying with 68.8% (preschool) and 37.8% (school children) of the dairy recommendations intake of the Food Based Dietary Guidelines according to ENCA [18].

The consumption of flour-based foods is widespread in the Chilean population. According to ENCA, 100% of Chileans consume bread daily (median 151 g/per day) [18], with this product being the principal source of sodium intake in the population (9.5 g per day) [1]. Evaluation of sodium content of the most consumed bread in Chile results in an average of 630.2 ± 112.0 mg/100 g [24]. Since 2010, the Chilean government has encouraged a voluntary reduction in the sodium content of bread, following international recommendations [25]. The positive results of this reformulation process [24,26] could influence the results of sodium content in flour-based foods in our baseline assessment (2013), however, over the study period, we observed a significant decrease in energy and sodium content in this food group.

Fats and oils presented a strong reduction in sodium content, but no changes were observed in saturated fat. This food group has a median consumption of 12.7 g-mL/day and 14.0 g-mL/day among adolescents and young adults, respectively, according to ENCA [18]. Decreasing the amount of saturated fat in this food group and others (e.g., confectionary and similar or desserts and ice creams) require changes in technological process and/or the use of other ingredients capable of maintaining sensorial aspects and food matrix structure, considering that increasing the amount of trans-saturated fatty acids in processed foods is not possible, as it is regulated in Chile since 2010 [8].

Sugary drinks presented a strong reduction in energy and total sugar content and these results could relate to two important Chilean regulations recently implemented: one that increased taxation of sugary drinks (2014) [27] and the Law 20.606 (2016) [8]. A recent study analyzed sugar-sweetened beverages purchased before implementation of Law 20.606 (2015) and after the first limit of implementation (2017), the authors found a decrease in the volume purchased of beverages with FoP “high in” label of 22.8 mL/capita/day (95% CI: −22.9 to −22.7; *p* < 0.001) and a consequent decline in calories purchased from these products of 11.9 kcal/capita/day (95%CI: −12.0 to −11.9; *p* < 0.001) [28].

Recent meta-analysis support warning labels on sugary drinks as a population-level strategy for reducing sugary drink purchases. Based on simulation studies, authors conclude that reducing sugary drink intake by as little as 15–30 calories per day could reduce obesity prevalence by 1.5% to 7.8% and type 2 diabetes prevalence by up to 6.8%, with effects on behavioral outcomes of smaller magnitude [29].

Interestingly, in some food groups, the absence of different product brands in 2019 compared with 2013 was pronounced (more than 20%). In the meat and derivatives group, this non-presence reached 41.4%, especially due to the absence of processed turkey meat products. The most important change in this food group was the decrease in sodium content in 13.0%. The meat food group is highly accepted by children and includes products like ham and sausages (hot-dog). While product reformulation supports a reduction in sodium intake, no decrease in saturated fat content was observed for this food group.

Although it is important to consider the significant changes in the nutrient content in the food groups, the presence or absence of FoP labels between the analyzed period becomes more relevant. For example, there was a reduction of products with FoP “high in”total sugar labels canned foods, dairy (liquid) and in sugary beverages groups in different proportions (−45.0%, −32.1%, −15.4% and −49.3%, respectively). These reductions suggest important changes in the availability of products and could impact in the population choices and consumption.

There are still many questions to be answered in terms of the effects of the Law 20.606. For example: how has reformulation affected the number of ingredients per product? has food consumption and nutritional status changed in Chile? are Chileans eating more natural foods since the implementation of the law? [20]. Some of these questions will take time to answer. For now, studies such as ours show that the food environment has changed and today there are fewer foods with FoP warning labels in some food groups like dairy (especially liquid products) and, as a result, consumers have more alternatives to purchase products with a lower levels of energy and nutrients of concern. On the other hand, these same products may be higher in artificial sweeteners and additives used to improve the sensorial characteristics of the products.

To date, Chile has conducted only one National Food Consumption Survey, which was carried out in 2009-2010 [18]. The behavior of purchasing and consuming food of the population can change and can be influenced by many factors like the food environment and advertising. Updated and representative data from population, considering this new scenario is necessary to further evaluate the impact of market changes in patterns of consumption.

During the initial implementation of the food law, no massive campaigns to discourage the consumption of foods with a FoP “high in” label was carried out. It is important to mention that healthy eating messages, based on Dietary Food Guidelines, that recommend increasing the consumption of fruits, vegetables, legumes, fish and low fat dairy products (in their natural state) and reducing the consumption of salt, sugar, and products with FoP “high in” warning labels, are part of counselling strategy used permanently in the primary health care system to promote healthy eating and discourage the consumption of products with FoP “high in” labels [30,31].

These results should be evaluated while considering the limitations of this study. One of these relates to the limited number of products evaluated, as we included only the most consumed products according to ENCA [18]. However, consumption patterns could have changed and it is possible that the selected and analyzed products in this study do not represent the same proportion of products currently consumed. Some food groups presented low sample, which may have negatively influenced statistical power to show changes. It is important to reinforce that our study design did not consider the replacement of the brand products, only the replacement of product type of the same brand. Another limitation refers to the fact that we simulated the presence or absence of FoP “high in” warning labels, without confirming the real presence of the label on the products.

The findings of this study could stimulate food industry to work on product reformulation and to develop more healthy products by decreasing the content of total sugar, saturated fat and sodium without the addition of chemical replacements. Understanding the impact of Law 20.606 can help in the planning and development of new policies to protect and improve healthy food environments [32,33]. Future studies should further analyze ingredient modifications of the most consumed products in the country.

## 5. Conclusions

We conclude that the Chilean Food Labeling and Advertising Law had an impact in the market and in the nutritional content of packaged foods. A great percentage of foods “high in” total sugar, saturated fat, sodium, and energy were withdrawn from the market. Also, a considerable number of products were reformulated, especially in the dairy (liquid), solid fats, sugary drinks, and flour-based food groups. The natural impact of this reformulation is the reduction in the expected number of products with a FoP “high in” label, which are exempt from the sales and advertising restrictions indicated by the law. It is important to mention that sugary beverages had a strong reduction in energy and total sugar content. This result may relate to the convergence of two recent regulations: the increase in taxation on sugary drinks (2014) and Law 20.606 (2016). It is necessary to continue developing public policies to protect and improve healthy food environments.

## Figures and Tables

**Figure 1 nutrients-12-02371-f001:**
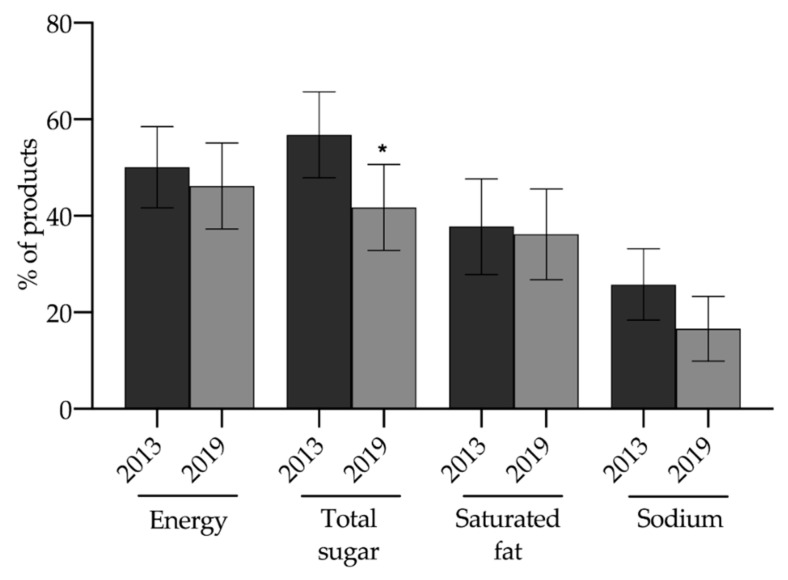
Proportion of packaged food (solid and liquids) meeting the criteria for FoP under Chilean nutrient profile limits by year. * *p* < 0.05.

**Table 1 nutrients-12-02371-t001:** Food groups according to Chilean Sanitary Regulation Food Code, year and losses observed in the period (2013–2019).

Food Group	2013	2019	Loss in the Period
	*n* (%)	*n* (%)	*n* (%)
Dairy	195 (35)	175 (37)	20 (10)
Flour-based food	66 (13)	58 (12)	8 (12)
Confectionery and similar	36 (6)	32 (7)	4 (11)
Confitures and similar	32 (6)	20 (4)	12 (37)
Fish and seafood	30 (5)	30 (6)	0 (0)
Sauces and dehydrated soups	29 (5)	29 (6)	0 (0)
Meat and derivatives	29 (5)	17 (4)	12 (41)
Sugary beverages	28 (5)	23 (5)	5 (18)
Cereals	23 (4)	19 (4)	4 (17)
Foods for special diet	17 (3)	15 (3)	2 (12)
Fats and oils	15 (3)	13 (3)	2 (13)
Desserts and ice creams	15 (3)	12 (2)	3 (20)
Spices, condiments and sauces	12 (2)	10 (2)	2 (17)
Sugars	11 (2)	11 (2)	0 (0)
Canned foods	9 (2)	8 (2)	1 (11)
Prepared meals	4 (1)	4 (1)	0 (0)
Total	551 (100)	476 (100)	75 (13.6)

*n*: number; %: proportion.

**Table 2 nutrients-12-02371-t002:** Description of energy and nutrients of concern (total sugars, saturated fats, sodium) declaration of solid food groups by year.

Food Group	Energy (kcal) Median (IQR)		Total Sugar (g) Median (IQR)		Saturated fat (g) Median (IQR)		Sodium (mg) Median (IQR)	
	2013	2019	2013	2019	2013	2019	2013	2019
Dairy	341 (224)	345 (236)	8 (35)	3 ^1^ (32)	15 (11)	16 (13)	360 (414)	375 ^1^ (353)
Flour-based food	375 (138)	358 ^1^(154)	11 (26)	6 (24)	1 (6)	1 (3)	289 (307)	272 ^1^ (269)
Confectionery and similar	484 (169)	487 (190)	55 (21)	53 (22)	18 (10)	17(6)	108 (153)	103 (88)
Confitures and similar	104 (19)	56 ^2^ (16)	25 (5)	10 ^2^ (1)	13 (0)	13 (0)	22 (10)	19 (8)
Fish and seafood	69 (79)	99 (58)	0 (0)	0 (0)	1 (1)	1 (1)	224 (187)	251 (205)
Sauces and dehydrated soups	352 (19)	349 (42)	8 (9)	9 (9)	2 (2)	1 (2)	5119 (1890)	4970(1965)
Meat and derivatives	197 (158)	164 (162)	0 (0)	0 (0)	5 (9)	2 (10)	752 (915)	850 ^1^ (782)
Cereals	377 (26)	374 (49)	23 (8)	18 ^2^ (16)	2 (2)	2 (3)	137 (160)	110 (180)
Food for special diet	80 (415)	79 (413)	2 (14)	12 (14)	1 (0)	0 (0)	86 (136)	50 (157)
Fats and oils	351 (27)	280 ^2^ (155)	1 (1)	0 (1)	14 (9)	10 (10)	657 (201)	389 ^2^ (183)
Spices, condiments and sauces	85 (63)	84 (49)	6 (2)	6 (16)	8 (1)	0 (0)	637 (940)	398 ^1^ (349)
Sugars	397 (18)	384 (31)	75 (43)	76 (55)	0.0 (6)	0 (0)	45 (125)	47 (62)
Canned foods	75 (10)	47 ^1^ (38)	18 (17)	6 (9)	0 (0)	0 (0)	156 (324)	116 (294)
Prepared meals	348 (40)	372 (76)	3 (3)	4 (1)	4 (3)	4 (3)	446 (485)	507 (470)

Wilcoxon signed-rank test; ^1^
*p* < 0.05; ^2^
*p* < 0.01; IQR: interquartile range; g: grams; mg: milligram.

**Table 3 nutrients-12-02371-t003:** Description of energy and nutrients of concern (total sugars, saturated fats, sodium) declaration of liquid food groups by year.

Food Group	Energy (kcal) Median (IQR)		Total Sugar (g) Median (IQR)		Saturated Fat (g) Median (IQR)		Sodium (mg) Median (IQR)	
	2013	2019	2013	2019	2013	2019	2013	2019
Dairy	81 (66)	67 ^2^ (61)	10 (9)	5 ^2^ (5)	1 (2)	1 (2)	62 (39)	58 ^2^ (20)
Sugary beverages	44 (45)	21 ^2^ (22)	11 (11)	4 ^2^ (3)	0 (0)	0 (0)	12 (8)	13 (8)
Fats and oils	828 (0)	828 (0)	0 (0)	0 (0)	15 (4)	14 (7)	0 (0)	0 (0)
Desserts and ice creams	101 (20)	94 (28)	13 (2)	13 (1)	2 (1)	2 (1)	40 (5)	56^1^ (12)

Wilcoxon signed-rank test; ^1^
*p* < 0.05; ^2^
*p* < 0.01; IQR: interquartile range; g: grams; mg: milligrams.

**Table 4 nutrients-12-02371-t004:** Proportion of simulated presence of FoP “high in” warning labels by solid food group products according to Chilean energy and nutrients of concern (total sugars, saturated fats, sodium) limits by year.

Food Group	Energy (kcal) % [95% CI] (*n*)		Total Sugar (g) % [95% CI] (*n*)		Saturated Fat (g) % [95% CI] (*n*)		Sodium (mg) % [95% CI] (*n*)	
	2013	2019	2013	2019	2013	2019	2013	2019
Dairy	50.0 [38.8–61.2] (38)	51.3 [40.0–62.5] (39)	27.4 [17.6–40.0] (17)	18.4 [11.1–29.0] (14)	69.8 [57.2–80.0] (44)	71.1 [59.6–80.3] (54)	34.2 [24.3–5.78] (26)	36.8 [26.6–48.4] (28)
Flour-based food	77.6 [64.8–86.7] (45)	63.8^1^[50.4–75.4] (37)	50.9 [37.9–63.8] (29)	46.6 ^1^ [33.9–59.7] (27)	31.5 [20.3–45.3] (17)	29.3 [18.8–42.6] (17)	29.3 [18.9–42.5] (17)	0 (0)
Confectionery and similar	81.3 [63.2–91.6] (26)	81.3 [63.2–91.6] (26)	81.3 [63.2–91.6] (26)	81.3 [63.2–91.6] (26)	81.8 [58.8–93.4] (18)	87.5 [69.8–95.5] (28)	6.3 [1.5–22.8] (2)	0.0 (0)
Confitures and similar	5.0 [0.6–31.1] (1)	5.0 [0.6–31.1] (1)	100.0 (20)	55.0 ^2^ [31.8–76.2] (11)	100 (1)	100 (1)	0.0 (0)	0.0 (0)
Fish and seafood	0.0 (0)	0.0 (0)	0.0 (0)	0.0 (0)	0.0 (0)	13.3 [4.8–32.0] (4)	16.7 [6.7–35.7] (5)	6.7 ^1^ [1.5– 24.7] (2)
Sauces and dehydrated soups	82.8 [63.3–93.0] (24)	79.3 [59.6–90.9] (23)	46.4 [28.2–65.7] (13)	44.8 [27.2–63.9] (13)	18.8 [5.3–48.6] (3)	6.9 [1.6–25.4] (2)	100 (29)	100 (29)
Meat and derivatives	23.5 [8.4–50.8] (4)	23.5 [8.4–50.8] (4)	0 (0)	12 [0–40.9] (2)	53.3 [27.4–77.6] (8)	35.3 ^1^ [15.2–62.3] (6)	52.9 [28.7–75.9] (9)	52.9 [28.7–75.9] (9)
Cereals	89.5 [63.9–97.6] (17)	89.5 [639–97.6] (17)	93.8 [62.4–99.3] (15)	79.0 [52.7–92.7] (15)	9.1 [1.0–50.9] (1)	10.5 [2.3–37.1] (2)	5.3 [0.6–32.5] (1)	5.3 [0.6–32.5] (1)
Food for special diet	6.6 [0.8–39.7] (1)	6.7 [0.8–39.7] (1)	36.4 [12.4–68.8] (4)	40.0 ^1^ [17.1–68.2] (6)	0.0 (0)	20.0 [5.6–51.2] (3)	0.0 (0)	0.0 (0)
Fats and oils	100.0 (6)	50.0 ^1^ [9.1–90.9] (3)	0.0 (0)	0.0 (0)	100.0 (6)	66.7 ^1^ [14.9–95.8] (4)	83.3 [23.0–98.8] (5)	33.3 ^2^ [4.2–85.1] (2)
Spices, condiments and sauces	20.0 [4.0–59.9] (2)	20.0 [4.0–59.9] (2)	22.3 [4.3–64.5] (2)	40.0 ^2^ [12.5–75.7] (4)	100.0 (2)	0.0 (0)	100.0 (12)	40.0 ^2^ [12.5–75.7] (4)
Sugars	36.4 [11.7–71.2] (4)	36.4 [11.7–71.2] (4)	44.4 [13.4–80.5] (4)	36.4 ^1^ [11.7–71.2] (4)	33.3 [0.7–99.7] (1)	0.0 (0)	0.0 (0)	0.0 (0)
Canned foods	0.0 (0)	0.0 (0)	57.1 [17.1–89.6] (4)	25.0 [4.1– 72.4] (2)	0.0 (0)	0.0 (0)	0.0 (0)	0.0 (0)
Prepared meals	75.0 [41.3–99.5] (3)	75.0 [41.3–99.5] (3)	0.0 (0)	0.0 (0)	33.3 [0.7–99.7] (1)	0.0 (0)	50.0 [24.7–97.5] (2)	75.0 ^1^ [41.3–99.5] (3)

Chi-square or Fisher Exact test ^1^
*p* < 0.05, ^2^
*p* < 0.01; *n*: number; %: proportion; CI: confidence interval; g: grams; mg: milligrams.

**Table 5 nutrients-12-02371-t005:** Proportion of simulated presence of FoP “high in” warning labels by liquid food group products according to Chilean energy and nutrients of concern (total sugars, saturated fats, sodium) limits by year.

Food Group	Energy (kcal) % [95% CI] (*n*)		Total Sugar (g) % [95% CI] (*n*)		Saturated Fat (g) % [95% CI] (*n*)		Sodium (mg) % [95% CI] (*n*)	
	2013	2019	2013	2019	2013	2019	2013	2019
Dairy	60.6 [50.6–69.8] (60)	46.5 ^1^ [36.7–56.5] (46)	75.0 [65.3–82.7] (72)	59.6 ^1^ [49.5–68.9] (59)	16.3 [9.6–26.2] (13)	22.5 ^1^ [14.9–32.5] (20)	6.1 [2.7–12.9] (6)	1.0 ^1^ [0.1–7.0] (1)
Sugary beverages	8.7 [2.0–30.6] (2)	4.3 ^1^ [0.6–28.4] (1)	66.7 [43.2–84.0] (14)	17.4 ^2^ [6.2–40.3] (4)	0.0 (0)	0.0 (0)	0.0 (0)	0.0 (0)
Fats and oils	14.3 [1.2–70.1] (1)	14.3 [1.2–70.1] (1)	0.0 (0)	0.0 (0)	14.3 [1.2–70.1] (1)	14.3 [1.2–70.1] (1)	0.0 (0)	0.0 (0)
Desserts and ice creams	100.0 (15)	100.0 (12)	100.0 (15)	100.0 (12)	27.3 [7.7–62.9] (3)	36.4 [11.7–71.2] (4)	0.0 (0)	0.0 (0)

Chi-square Test or Fisher Exact ^1^
*p* < 0.01; *n*: number; %: proportion; CI: confidence interval; g: grams; mg: milligrams.

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
