# Peer review of "Changes in Nutrient Declaration after the Food Labeling and Advertising Law in Chile: A Longitudinal Approach"

_nutrients, 2020, doi:10.3390/nu12082371_

Round 1
Reviewer 1 Report
Greetings,
This is an important study on the impact of food labeling policy on the supply side of food consumption. The researchers have gone through an exhaustive sampling and evaluation of shelf products and performed detailed analysis of food items composition (package labels) before- and after- the implementation of the law.
The study has high value in the context of public health policy and regulations and has a robust methodology given the limitations of observational studies. I would like to point out to a few concerns and suggestions with regards to methodology, and mainly the interpretation of the results:
1) the authors mention that they removed 100 items from original list (650 items) because of quality of the pictures available, but furthermore, there was ~13.6% loss in 2019 sample collection due to discontinuation of product, etc. It is not clearly mentioned in the statistical methods whether or not these items were removed from the analysis. How this loss would impact generalizability of the changes in each food group?
2) The authors mention that they have tested the nutrition properties of the missing items and they were generally similar to the rest of items in their respective group, but given that in some groups this missingness is proportionally high (confitures; ice-cream in addition to meat and its derivatives) this limitation deserves further elaboration. For instance, there was little change detected in ice cream and pastry groups, is it possible that the study did not have enough power to show changes in this group; 20% loss in original sample.
Noting that the authors mention this limitation with regards to meat products.
3) Presumably there are new products entering the market during the 6-year period from the baseline. Although this is not within the scope of the study, reformulated new products may present a different landscape for nutritional value of what really constitutes 70% of purchased products in 2019. The fact that the dietary recall pertains to the baseline, the generalizability of findings should be cautioned because of possible changes in consumer basket over this period (overall, or in certain food groups). For instance, in the statement in line #350 "today there are fewer foods with FoP warning labels in most food groups".
Again, the authors have raised this issue in the discussion but merely with regards to meat products. In order for these findings to be applied in policy decision, this issue should be taken into consideration.
4) The authors very well mention the potential confounding effect of other policies, (voluntary reduction of sodium content in bread, taxation on sugary beverages) and the delayed effect of policy on certain products that need more complex reformulation and major alterations in the technological processes. While these aspects are very well described, less is discussed about consumer behavior and possible campaigns that may further affect 'consumer demand', which in turn, may have contributed to discontinuation or reformulation/rebranding of certain products.
Since the campaign to fight overweight/obesity and diet-related NCD is multi-faceted, there might be several pathways in play to change consumer behavior and therefore, the market demand.
Overall, the manuscript provides timely evaluation of policy to improve nutritional value of food products in the market; it uses a sound methodology and the findings are presented in a clear and systematic manner. I have minor suggestions to clarify the interpretations and conclusions so that a potentially broader audience (specifically non-scientist policy makers) would understand the implications of findings.
Author Response
REVIEWER #1 - Comments and Suggestions for Authors
- The authors mention that they removed 100 items from original list (650 items) because of quality of the pictures available, but furthermore, there was ~13.6% loss in 2019 sample collection due to discontinuation of product, etc. It is not clearly mentioned in the statistical methods whether or not these items were removed from the analysis. How this loss would impact generalizability of the changes in each food group?
Answer: We thank the reviewer for this comment. Despite extensive training, pictures from about 100 products did not have all the necessary information (nutrient and ingredient list) or pictures were blurred and, thus, were eliminated from the database. This elimination happened in the quality analysis phase, in 2013, where independent coders evaluated the collected pictures.
In relation to the 13.6 %, this percentage represents “loss” in the 6-years period, i.e., products that despite an exhaustive search in different supermarket chains and local stores were not found in the stores in 2019. These products, initially, were not excluded from the database, since they did not present statistical differences in the nutrient profile, nor in the FOP simulations (p>0.05). However, after considering reviewer suggestions, all analyses were revised to exclude these products. As a result, all tables (2,3,4, and 5) now just consider the 476 products. This is explained in lines 167-174.
- The authors mention that they have tested the nutrition properties of the missing items and they were generally similar to the rest of items in their respective group, but given that in some groups this missingness is proportionally high (confitures; ice-cream in addition to meat and its derivatives) this limitation deserves further elaboration. For instance, there was little change detected in ice cream and pastry groups, is it possible that the study did not have enough power to show changes in this group; 20% loss in original sample. Nothing that the authors mention this limitation with regards to meat products.
Answer: We appreciate the suggestion to carefully detail the limitations of our study. We have included in the limitations section an observation about the possible relationship between food losses and statistical power (see lines 387-388).
- Presumably there are new products entering the market during the 6-year period from the baseline. Although this is not within the scope of the study, reformulated new products may present a different landscape for nutritional value of what really constitutes 70% of purchased products in 2019. The fact that the dietary recall pertains to the baseline, the generalizability of findings should be cautioned because of possible changes in consumer basket over this period (overall, or in certain food groups). For instance, in the statement in line #350 "today there are fewer foods with FoP warning labels in most food groups". Again, the authors have raised this issue in the discussion but merely with regards to meat products.
Answer: We appreciate this comment and agree with the reviewer on this point. In 6 years, many changes in product availability may occur. Unfortunately, we do not have another Food Consumption National Survey to corroborate that the most consumed foods are the same or have changed. The methodology section details that information from 2013 was used. Additional information regarding this point was added to the discussion Lines 371-375 and 385-387.
- The authors very well mention the potential confounding effect of other policies, (voluntary reduction of sodium content in bread, taxation on sugary beverages) and the delayed effect of policy on certain products that need more complex reformulation and major alterations in the technological processes. While these aspects are very well described, less is discussed about consumer behavior and possible campaigns that may further affect 'consumer demand', which in turn, may have contributed to discontinuation or reformulation/rebranding of certain products. Since the campaign to fight overweight/obesity and diet-related NCD is multi-faceted, there might be several pathways in play to change consumer behavior and therefore, the market demand.
Answer: We appreciate this comment and agree that consumer demand may encourage changes in product availability. However, in Chile, no large-scale educational campaign focused on reducing the consumption of foods with FoP warning labels. A very short “campaign” was carried out in 2016 and related to the implementation of the law to inform consumers about FoP labels on food packages. The law required that products with FoP high-in labels had to add the following messages to advertisements “prefer food with less FoP high-in labels”. Some food companies voluntarily added this same message to food packaging. This information was added to the discussion lines 376-382.

Reviewer 2 Report
Overall the paper contains results of interest to policy makers, the food industry and academics. The findings are important and useful, as they demonstrate they demonstrate, with in the limitations of the study design, that the introduction of Front of Pack (FoP) warning labels in Chile was associated with a reduction of a range of nutrients in foods and drinks. Other countries will find this useful in developing and taking forward labelling schemes.
The paper is however quite difficult to read. Sentence constructed, and the use of some words and phrases, make it hard to make sense of at times. It would therefore encourage, that is subject to a strong edit improve English, clarity and understandability.
This finding, were associated with a range of other interventions in Chile including advertising restrictions, these may have also been contributed to the product changes observed. I suggest that this is reflected up in the conclusion. The limitations of the study also need to be more clearly addressed – products were only collected from one super markets. It is not clear what proportion of products that represented.
Specific points:
Line 23 – clarify methodology and sampling
Line 26- don’t say ‘little change’. Give the % change as it is significant
Throughout – the phrase ‘critical nutrients’ is used. I found this jarring, as it to close in meaning to ‘essential’ nutrients which has a precise and very different nutritional meaning.
Line 41-45- I think the WHO has policies have recommendations about reducing consumption of foods high in salt, fat, sugar etc. I don’t remember them m specially talking about process or ultra-processed foods. Suggest checking the accuracy.
Line 46-47 – the description of the relationship of the polices to the nutrient profile was hard to follow. I suggest simplifying to say something like – only products not high in calories, total sugar, saturated fat and sodium as measured by a nutrient profile (ref) could be advertised or /and avoid a FoP warning label.
Methods – add in % products sampled out of those available within each food group, and what proportion of the market the supermarket used represents
Food classification – I don’t see the usefulness of including foods that cannot be reformulated such as sugar and fish. I suggest these are removed
Line 2001 -2 – add P vales
Fig 1 – add P values
The tables - all need attention as they are not aligned – making them hard to read
Tables 2- remove decimal places for energy and sodium – they imply a precision that isn’t there
Line 227 – the reduction in sodium is meaningfully, so I would just sat a 5% reduction
Line 280 – ‘according to food companies---’ suggest saying more about why the said FoP didn’t lead to much reformulation
Line 322 - I would be surprised if oils and fats are largely consumed by adolescents. Think this is an English issue.
Line 340-345 – I would suggest that if a large proportion of products cannot be matched in the 2 data sets, them they should be removed from the analysis as any apparent changes nutrient levels may just be down a bias introduced by this.
Conclusions need editing
Author Response
Reviewer #2 - Comments and Suggestions for Authors
- The paper is however quite difficult to read. Sentence constructed, and the use of some words and phrases, make it hard to make sense of at times. It would therefore encourage, that is subject to a strong edit improve English, clarity and understandability.
Answer: A native English speaker has carefully reviewed the article.
- This finding was associated with a range of other interventions in Chile including advertising restrictions, these may have also been contributed to the product changes observed. I suggest that this is reflected up in the conclusion. The limitations of the study also need to be more clearly addressed – products were only collected from one super markets. It is not clear what proportion of products that represented.
Answer: We have carefully revised the limitations section of the article. Additionally, we want to comment to the reviewer that the same law (20,606) restricts the advertising of “high-in” products for children under 14 years old. This law complements another law (20,869), implemented in 2018, that restricts the advertising of “high-in” products between 6 am and 10pm on television and in movie theaters. Supermarket data collection occurred in the biggest supermarket chain in the country (not a single supermarket). After an exhaustive search in other supermarket chains and local stores, some products were not found. The objective of this study was not to demonstrate the availability of products in the market/retail space, but rather to evaluate the most consumed foods (70%) as defined by the only Food Consumption National Survey carried out in 2009-2010. Thus, all these products were searched in retail spaces. The limitations were better described see lines 383-391.
- Line 23 – clarify methodology and sampling
Answer: Included more details see lines 20-23.
- Line 26- don’t say ‘little change’. Give the % change as it is significant
Answer: This section was entire restructured, see lines 23-28.
- Throughout – the phrase ‘critical nutrients’ is I found this jarring, as it to close in meaning to ‘essential’ nutrients which has a precise and very different nutritional meaning.
Answer: We thank the reviewer for this comment and agree. Despite the widespread use of this concept in many publications, we decided to replace term with “energy and nutrients of concern (total sugar, saturated fats, sodium)”. This change was made throughout the article.
- Line 41-45- I think the WHO has policies have recommendations about reducing consumption of foods high in salt, fat, sugar etc. I don’t remember them specially talking about process or ultra-processed foods. Suggest checking the accuracy.
Answer: We have revised all documents to reflect that this concept is referred to in publications of the PAHO. We have changed the term “ultra-processed” to “unhealthy foods” (see line 43).
- Line 46-47 – the description of the relationship of the polices to the nutrient profile was hard to follow. I suggest simplifying to say something like – only products not high in calories, total sugar, saturated fat and sodium as measured by a nutrient profile (ref) could be advertised or /and avoid a FoP warning label.
Answer: We have modified this sentence (see lines 46-59).
- Methods – add in % products sampled out of those available within each food group, and what proportion of the market the supermarket used represents
Answer: Additional information was provided (see line 106). However, as previously mentioned, the purpose of this study was not to cover a certain percentage of available products or food groups in the supermarket, but rather to evaluate the most consumed foods.
- Food classification – I don’t see the usefulness of including foods that cannot be reformulated such as sugar and fish. I suggest these are removed
Answer: We understand the suggestion, and we want to clarify to the reviewer that all the included products in simulation of FoP was possible to be reformulated. In relation to sugar, that is included in sugars group, it was naturally excluded from the simulation FoP as well oils, wheat flour, raw oat, etc. This is defined by the fifth article(§5) of Law 20,606 (http://bcn.cl/2eu90) and Chilean Sanitary Food Regalement article 120 bis (https://dipol.minsal.cl/wp-content/uploads/2019/06/DECRETO_977_96_2019-2.pdf ) that mention the FoP High-in is just inserted in the packaged foods added of sugars, saturated fat and sodium when the final nutrient content exceeds the limit stablished by the Ministry of Health.
In the case of fish, included in the fish and seafood group, it was considered just the canned ones. Raw products were not considered in this study. So, after this brief explanation we hope the reviewer understand that it is not appropriated to exclude these foods. Furthermore, the canned fish is the type of fish most consumed by low socioeconomic level population.
We revised the explanations about that and was presented in the section “2.3 Simulation of “high-in” FoP warning label”, see lines 142-150.
- Line 2001 -2 – add P vales
Answer: Included in line 182.
- Fig 1 – add P values
Answer: Included.
- The tables - all need attention as they are not aligned – making them hard to read
Answer: Revised as requested.
- Tables 2- remove decimal places for energy and sodium – they imply a precision that isn’t there
Answer: Revised as requested (tables 2 and 3).
- Line 227 – the reduction in sodium is meaningfully, so I would just sat a 5% reduction
Answer: We have analyzed all the data again and new p values was calculated.
- Line 280 – ‘according to food companies---’ suggest saying more about why the said FoP didn’t lead to much reformulation
Answer: We would ask the reviewer to confirm the line number, as line 280 does not seem to relate to the comment provided. The reviewer may be referring to line 307 that discusses product reformulation. This sentence refers to a Ministry of Health report (ref 21) that details the progress of law implementation after the first stage (2016). The report states that food companies indicated that 18% of products underwent reformulation. Since that report, no other publicly available information in relation to product reformulation has been made available.
- Line 322 - I would be surprised if oils and fats are largely consumed by adolescents. Think this is an English issue.
Answer: We understand that this may have surprised the reviewer, but it was not a misunderstanding. This information can be found in the final report of the National Food Consumption Survey (ENCA) (ref18) which indicates that, compared to other groups of the population, fats and oils are consumed in high quantities among adolescents and young adults in Chile. We have revised the sentence for clarity (see lines 330-332).
- Line 340-345 – I would suggest that if a large proportion of products cannot be matched in the 2 data sets, them they should be removed from the analysis as any apparent changes nutrient levels may just be down a bias introduced by this.
Answer: We thank the reviewer for this suggestion. Despite the provided information that there was not a significant difference in the nutrient profile of these 75 products, as a result of reviewer comments, we decided to remove these products from the database. All analyses were revised and now reflect the same 476 products available in 2013 and in 2019.
- Conclusions need editing
Answer: Revised as requested, see lines 401-410.
